# Experiences with Family Planning amongst Persons with Mental Health Problems: A Nationwide Patient Survey

**DOI:** 10.3390/ijerph20043070

**Published:** 2023-02-09

**Authors:** Noralie N. Schonewille, Monique J. M. van den Eijnden, Nini H. Jonkman, Anne A. M. W. van Kempen, Maria G. van Pampus, Francisca G. Goedhart, Odile A. van den Heuvel, Birit F. P. Broekman

**Affiliations:** 1Department of Psychiatry and Medical Psychology, OLVG, Oosterpark 9, 1091 AC Amsterdam, The Netherlands; 2Department of Psychiatry, Amsterdam UMC, Vrije Universiteit Amsterdam, Boelelaan 1117, 1081 HV Amsterdam, The Netherlands; 3Amsterdam Public Health, Mental Health Program, Amsterdam UMC, Vrije Universiteit Amsterdam, Boelelaan 1117, 1081 HV Amsterdam, The Netherlands; 4Team Knowledge, Innovation and Research, MIND, Stationsplein 125, 3818 LE Amersfoort, The Netherlands; 5Department of Research and Epidemiology, OLVG, Oosterpark 9, 1091 AC Amsterdam, The Netherlands; 6Department of Neonatology and Pediatrics, OLVG, Oosterpark 9, 1091 AC Amsterdam, The Netherlands; 7Department of Gynecology and Obstetrics, OLVG, Oosterpark 9, 1091 AC Amsterdam, The Netherlands; 8Department of Anatomy & Neuroscience, Amsterdam UMC, Vrije Universiteit Amsterdam, Boelelaan 1117, 1081 HV Amsterdam, The Netherlands; 9Amsterdam Neuroscience, Compulsivity, Impulsivity & Attention Program, 1081 HV Amsterdam, The Netherlands

**Keywords:** family planning, psychiatry, mental health, childlessness, sexuality, parenting, taboo, unintended pregnancies

## Abstract

High rates of unintended pregnancies in patients with mental health problems reflect the unmet need for tailored family planning. This study aims to explore aspects of family planning that are especially challenging for patients experiencing health problems by obtaining the perspective of (former) patients and those with close relationships with the (former) patients. In August 2021, members of a Dutch national mental health panel, consisting of (former) patients and close ones, were invited to respond to a 34-question online survey that included questions on four domains: reproductive history, decision making, parenting, and sexuality. This study has revealed the severe and adverse impact of mental health problems across all of the four domains of reproductive health and family planning, which the questions specifically targeted. Based on these results, we recommend discussing family planning with all patients experiencing or at risk for mental health problems and their partners. These discussions should address a desire to have children, (involuntary) childlessness, uncertainties about parenting and sexuality, while remaining considerate of experienced taboos.

## 1. Introduction

Family planning is an important aspect for most people at some point in their lives [1]. The working definition of family planning (‘the ability of individuals and couples to anticipate and attain their desired number of children and the spacing and timing of their births’, used by the WHO Department of Reproductive Health and Research [2008]), includes prevention of unwanted pregnancies as well as obtaining desired pregnancies. Researchers have become increasingly interested in studying the risk factors for impaired family planning, as unwanted pregnancies have adverse impacts on maternal and child health [2].

One of these risk factors is impaired mental health, as mental health problems can coincide with several aspects of family planning.

Unintended pregnancies are often associated with misuse or non-use of contraceptive methods [3,4]. For women with severe mental illnesses such as schizophrenia and bipolar disorders, efficient use of contraceptive methods might, at times, be challenging. This arises from impaired decision-making and advanced planning skills during disease episodes [3,4,5,6]. Aside from proper contraceptive use, eating disorders or use of antipsychotic medication disrupted menstruation and could lead to an incorrect belief with respect to fertility [7,8].

Although impaired mental health is an established risk factor for unintended pregnancies [9], the perspective and preferences of persons with mental health problems on their own family planning are currently lacking. Moreover, most studies have a profound focus on preventing unwanted pregnancies and only a minor interest in studying how desired pregnancies can be achieved. As the goal of family planning is to have reproductive intentions met, involuntary childlessness amongst persons with mental health problems should be included in the study of family planning. Several factors related to mental health problems could interfere with achieving desired pregnancies, such as problems with sexual functioning [10], concerns about passing on heritable psychiatric conditions [11], and fears of ‘not being a good parent’ [12]. Previous literature has shown that persons with mental illnesses (schizophrenia, autism, eating disorders, substance abuse, and/or depression) have lower fecundity when compared to their unaffected siblings [13,14]. These population-based studies clearly suggest that persons with mental health problems might struggle to have their reproductive intentions met.

Despite these additional challenges facing mental health patients in relation to family planning, their perspectives on childlessness have received little scientific attention.

A few qualitative interview studies have discussed family planning in small samples of women with severe mental illness [11,15]. These studies demonstrate a lack of findings from large samples of patients sharing experiences with family planning in relation to various mental health problems, such as depression, anxiety, traumatic experiences, or personality disorders. Additionally, the view of those with close relationships to the patient, i.e., spouses or broader partners, family members, or close friends (from now on named ‘close ones’) on family planning is generally not included in studies.

In this study, we aimed to elucidate the perspective on family planning from patients with mental health problems and their close ones. We hypothesize that patients and close ones experience various difficulties regarding family planning and subsequently have the desire to discuss family planning with mental health professionals. Understanding the patient’s perspective will help to tailor family planning counseling to the needs of patients with mental health problems and their close ones.

## 2. Materials and Methods

The current study reports the results from a survey that was electronically dispersed in June 2022 among a panel of (former) patients and close ones (the MIND mental health care panel) and was available for four weeks. MIND is a Dutch association for former or current patients with mental health problems and close ones of (former) patients with mental health problems. MIND represents the whole spectrum of mental health (www.wijzijnmind.nl (accessed on 1 December 2022)). Members of the MIND panel are recruited among the public with a request for participation ‘’Do you have experience with mental health problems yourself or as a close one?’’. There are no other selection criteria. The panel consists of 4200 (former) patients and close ones. The members of the panel are invited to complete surveys through an email invitation approximately 10 times a year.

The current survey was developed in close collaboration with the association of patient and relative organizations MIND and a patient–investigator, who is part of the research team (ME). In addition, an expert panel consisting of members of MIND, in addition to a pediatrician, an infant mental health specialist, a gynecologist, a child psychiatrist, an epidemiologist, and two psychiatrists with experience in the field of mental health and family planning, were consulted. The 34 questions, provided in the Dutch language, were a combination of open questions with free-form text fields, multiple choice questions, and 5-point Likert scales. The estimated time to populate the questionnaire was 10–15 min. The first question category regarded patient histories of mental health problems (both diagnosed and self-reported problems), psychoactive medication use, pregnancies including elective abortion and unintended pregnancies, and childlessness. The second question category regarded the perceived relation between mental health and family planning, the experience with the conversation about family planning in mental health care, and the perception of stigmas and taboos regarding mental health problems, as well as sexuality in both mental health care and the personal environment. Close ones were identified by a first question ‘Do you fill in this survey as a patient or as a close one (of a person with mental health problems)?’. For those questionnaires populated by close ones, they received similar questions as (former) patients that specifically asked about the experiences of their close ones. For the current paper, the survey was translated into English in collaboration with a native speaker.

This study included all panel members who populated the survey and provided informed consent. Respondents under the age of 18 years were excluded (as derived from the survey responses).

Anonymized data were collected in the software program Spidox (www.spidox.net (accessed on 1 December 2022)) and shared with the research team in a secured Excel file. Subsequently, data were imported into R studio (version 4.2.0) for data cleaning and analysis. We performed descriptive analyses of the respondents’ demographics and history of mental health problems. Age was presented as mean and standard deviation (in the case of normal distribution). Other descriptive characteristics were presented as numbers and proportions of the group (all, female, men, other gender, or close ones). Living area was assessed with the question ‘What is your living area? Rural/urban’. Education levels were assessed according to the International Standard Classification of Education (ISCED) levels 0–8. Mental health disorder was assessed with the question ‘Have you ever received a psychiatric diagnose? Yes/no’. Mental health problems were assessed with the question ‘What kind of mental health problems have you encountered in your life? Open question’. Recovery status was assessed with the question ‘Do you currently consider yourself recovered? Yes/no/I have learned to live with it/other’. To present histories of mental health problems in the results section, we grouped self-reported mental health problems into larger categories (for example, depression, anxiety, etc.). Grouping was performed by a medical doctor and a researcher in psychology (NS and BW). A Cohen’s Kappa was used to measure interrater reliability for grouping between the two researchers [16].

The study group paired questions on similar topics into four domains, reproductive history, reproductive decision making, parenting, and sexuality. Reproductive history was reported separately for respondents identifying as women and men. History of abortion was presented for respondents identifying as women. The results of multiple-choice questions were presented in graphs. Open-text answers that elaborated on multiple choice answers (‘if yes, why?’) were collected and included in the results’ section to illustrate the findings. Respondents’ answers were not cited, nor did we calculate the number of reasons provided by the respondents.

Outcomes were presented for two groups separately: a group of respondents who populated the questionnaire as (former) patients and a group of close ones who represent the views of a close relation experiencing mental health problems. Additionally, we stratified responses for (former) patients and close ones with and without children. Confirmative responses were compared between men versus women or (former) patients with versus without children by Chi^2^ tests with a *p*-value considered significant at <0.05.

## 3. Results

The questionnaire was populated by 381 panel members (response rate of 9%). One respondent was excluded due to age <18 years, and two respondents failed to provide informed consent. Data from 378 respondents (354 (ex-)patients and 24 close ones) were included in the analyses (see Table 1 for sample characteristics). Close ones were family members (*n* = 18), partners (*n* = 4), or friends (*n* = 2). Respondents were mostly women (81.2%); nearly half of them were urban living (54.5%) and higher educated (50.2%). Fifty-one respondents were men. Most respondents were diagnosed with ≥1 mental health problems (95.5%), and (history of) symptoms of depression (58.7%) were most frequently reported. The inter-rater reliability for the classification of self-reported mental health problems was κ = 0.70 (*p* < 0.001), indicating a good level of agreement between the assessors [17,18].

### 3.1. Domain 1: Reproductive History

The reproductive history of women and men is presented in Figure 1. Appendix A provides all raw data supporting Figure 1. The proportion of unintended pregnancies in this sample was 21.7%. Unintended pregnancies occurred in almost half of all respondents (45.4%) who were ever pregnant (or the biological father of an unintended pregnancy in the case of men) (data not presented). Amongst the (former) patients, women had significantly more pregnancies (47.8% versus 11.8%, *p* < 0.001) and children (46.4% versus 11.8%, *p* < 0.001) compared to men (see Figure 1). Respondents elaborated on the relationship between unintended pregnancies and mental health problems with examples: alcoholism led to the contraceptive method being unreliable and attention deficit attributed to an unintended pregnancy through obliviousness. The proportion of elective abortion was 5.4%. A total of 10.6% of those who were ever pregnant have had at least one abortion.

### 3.2. Domain 2: Reproductive Decision Making

Experiences with the reproductive decision-making of (former) patients and close ones are presented in Figure 2. Appendix A provides all raw data supporting Figure 2. Reproductive decision-making was influenced by mental health problems in at least 25.7% of the (former) patients (Figure 2b, question 5). Several factors were mentioned where mental health problems influenced (the desire for) having one or more children: the belief that pregnancy is only possible after being recovered from mental health problems, the belief that physical violence in history made pregnancy impossible, and difficulty sustaining a partner relationship led to the belief that pregnancy was impossible. In addition, the severity of the mental health problems at certain points in time had an influence on the desire for a child. Having a stable mental health outlook, having financial stability, and being able to raise children were mentioned as factors that were a prerequisite for becoming pregnant.

The planning of pregnancies was influenced by mental health problems in many ways. Respondents reported that they chose to postpone a possible pregnancy until the point they felt mentally stable themselves or had a stable relationship with a partner. The fear of becoming mentally unwell during the postpartum period and fear for the health of the baby caused respondents to postpone their wish for pregnancy. Decreasing the dosage of medication or quitting medication was often reported as a prerequisite to becoming pregnant. Medication use was also reported to coincide with having a safe pregnancy.

Many close ones also recognized the relationship between mental health problems and various reproductive decisions, except for the question on experiencing taboo (Figure 2, question 7). While a taboo on discussing mental health problems was experienced by most (former) patients (62.4%), this was less mentioned by close ones (25.0%).

(Former) patients explained that there was no space to discuss their problems amidst the other problems in traumatized families and that talking about mental health problems was not deemed to be ‘appropriate’. (Former) patients described feeling perceived as ‘crazy’, weak, and felt they should not complain. Moreover, (former) patients reported feeling ashamed, experiencing denial, and feeling misunderstood by family, co-workers, or community. Of the (former) patients and close ones who desired to discuss family planning, only one in five had discussed it with a mental health professional. A subgroup analysis showed that respondents <40 years of age more often spoke about family planning than respondents >40 years of age (Figure 2b, question 7).

Of the childless respondents (*n* = 207), 48.3% attributed childlessness to their mental health state (Appendix A). After stratification for childlessness, (former) patients without children confirmed the influence of their mental health problems on the desire to have children (75.4% versus 27.2%, *p* < 0.001) and the ability to become pregnant (36.7% versus 17.0%, *p* < 0.001) more often than (former) patients without children. Almost half of all childless (former) patients never had a conversation about family planning (45.9%). In free-form text boxes, respondents further elaborated that childlessness was related to the fear of transmitting one’s mental health disorder or (childhood) trauma to their children, which led them to refrain from having children. Additionally, respondents described a feeling of responsibility in preventing a child from any harm (as they had often endured themselves). It was a conscious decision to be childless for some (couples), while others that had experienced mental health problems (or the treatment thereof with psychoactive medication) attributed this to be the source of fertility problems that resulted in childlessness. Lastly, respondents experienced discouragement from others to have children. Respondents reported that they received no tailored help. Relief and grief about childlessness were both mentioned.

### 3.3. Domain 3: Parenting

In total, 151 respondents (4 close ones and 147 (former) patients had children or reported for a person with children (in the case of close ones)). Figure 3 displays views on the influence of mental health on parenting and early parenthood. Appendix A provides all raw data supporting Figure 3.

One-third of 3% of (former) patients expressed that their mental health situation was related to all aspects of parenthood. Approximately half of them felt supported by their partners during parenthood and when mental health impacted the lives of their children (46.9% and 49.0%, respectively).

Close ones in our sample responded differently: questions regarding the impact of mental health problems during pregnancy and in the first years of parenthood, in addition to feeling impaired as a parent, were answered with ‘yes’ by close ones to a lesser extent (respectively, 0.0% and 25.0%). Partner support was present according to 75.0% of close ones in our sample, and support when mental health problems impaired the lives of children was present according to all responding close ones 100% of the time.

In free-form text boxes, (former) patients explained that a history of mental health problems also aided in their parenting. The ability to understand their child’s needs as they often resembled their own needs, the ability to reflect on one’s own actions because of trained self-reflection in therapy, and the ability to ask for help in a timely manner were mentioned as positive aspects of (a history of) mental health problems.

### 3.4. Domain 4: Sexuality

Figure 4 presents results for sexuality and contraceptive use. Appendix A provides all raw data supporting Figure 4. Most of the (former) patients experienced a taboo in discussing sexuality (in general), which was not reflected in the close ones’ view. In free-form text boxes, (former) patients explained that conversation about sexuality was not permitted in previous generations, in addition to experiences of abuse (in the nuclear family) and feelings of shame. Respondents also mentioned that mental health problems caused by (physical) trauma interfered with sexuality. Moreover, (former) patients explained that not being able to talk about sexuality worsened sexual problems, and both (former) patients and close ones agreed on mental health problems negatively influencing sexual enjoyment.

Contraceptive use when at risk for pregnancy (i.e., being sexually active without intention to become pregnant) was the norm (83.3%) according to the close ones and (former) patients (75.3%). Reasons reported by (former) patients and close ones for not using contraceptive methods were: being in a same-sex relationship, having an involuntary sexual relationship, being convinced of one’s infertility due to long periods of amenorrhea, having an infertile partner, being convinced by a partner to not use contraceptive methods, not being adherent to contraceptive pills, using other medication that influenced contraceptive effectiveness, and being intolerant to the use of hormonal contraceptive pills. Stratifications for (former) patients with (*n* = 147) and without children (*n* = 207) showed that experiences of taboos and sexual enjoyment were comparable (*p* = 0.506 and *p* = 0.971, respectively) (Appendix A).

## 4. Discussion

### 4.1. Key Results

Our results, derived from survey data from a panel of (former) patients with mental health problems and close ones, reflect the severe and adverse impact of mental health problems on four domains of reproductive health and family planning: reproductive history (including unintended pregnancies), reproductive decision making, parenting, and sexuality. High proportions of unintended pregnancies (45.4%), childlessness related to mental health problems (48.3%), experiencing taboo around sexuality (58.2%), and feeling impaired as a parent due to mental health problems (59.9%) illustrate common challenges that (former) patients with mental health problems face. Given these findings, it is striking that only one in five (former) patients who desired to have a conversation about family planning with a mental health professional were able to have that discussion.

### 4.2. Interpretation of Findings

Amidst the (former) patients, the proportion of unintended pregnancies (21.7%) was comparable to the lifetime prevalence of unintended pregnancy in the general Dutch population (20.0%) [19]. However, the proportion of unintended pregnancies amongst women who were ever pregnant in our sample was comparable to other samples of women with anxiety and depression (45.4%) [20]. This confirms the high risk of unintended pregnancies in persons with mental health problems, as previously shown in a review and meta-analysis [9]. Moreover, as a proportion of women in our sample is in the reproductive phase of life, the lifetime prevalence of unintended pregnancies could increase in our sample.

Contraceptive use through self-reporting was high in our study (75.3% according to (former) patients and 83.3% according to close ones). The discrepancy between contraceptive use and unintended pregnancies is generally explained by the suboptimal use of contraceptive methods. This is a common pitfall and appears to exist irrespective of educational level [21]. However, based on the results from our survey, additional reasons have been uncovered, such as intolerance to contraceptive methods and the perception of being infertile (due to mental health problems). Previous studies reported that unintended pregnancies were related to decreased sexual autonomy, impaired coping mechanisms (intimacy-related, ability to say no, or to ask for contraception), involvement in violent relationships, lack of knowledge about unintended pregnancies, or difficult access to, or interaction with, contraceptives [15,22]. In line with these findings, respondents of our study illustrated the inability to protect against unintended pregnancy with examples of involuntary intercourse, forced non-use of contraceptive methods, and interaction of contraceptive methods and other medication.

Interestingly, 48.3% of the (former) patients in this panel attributed their childlessness to their mental health. For 8.2% of childless (former) patients, mental health was not related to their childlessness. This is in concordance with a Scandinavian population register study that found associations between several major diseases, including mental disorders and being childless [13]. We found that insecurities about being a good parent could be attributed to (involuntary) childlessness, as explained by respondents. Close ones in this sample less frequently reported that mental health problems impaired the mental health patient as a parent or that mental health problems were related to pregnancies or early parenthood. This illustrates how the uncertainties that people with mental health problems encounter are not always visible or understood by others. Our data confirm prior research: fear of transmission of disease increases the fear of becoming a parent [23]. In addition, an unmet need to discuss family planning with mental health professionals could feed beliefs about harmful medication, infertility, and the inability to be a good parent, as described by the respondents. We did not specifically inquire about unwanted childlessness in this study, and thus have no knowledge about the proportions of respondents feeling regret, grief, or satisfaction with having no children and the relationship with mental health problems. Future (qualitative) studies could pay attention to childlessness in relation to mental health.

(Former) patients in this sample experienced a taboo in talking about family planning and sexuality in the context of their mental health problems. These experiences were hardly recognized by close ones. Furthermore, (former) patients felt supported in their mental health problems to a lesser extent than close ones indicated. Feeling an absence of support can negatively impact mental health, as unsupportive responses to mental health problems make it less likely that patients would seek help, as illustrated in another sample of patients with mental health problems and their close ones [24]. In addition, fear of being stigmatized added to the reluctance of patients to express their worries and needs to close ones and mental health professionals [24,25]. This might explain the reported unmet need for discussing family planning in our study. Previous research has shown that mental health professionals considered taboo and stigma barriers to providing care for their patients [26]. For patients with a mental health disorder, complex care needs, care avoidance, and lack of trust in mental health professionals hindered patients from seeking help. This can be further compounded by the practical aspects of healthcare, such as lack of time and waiting lists. However, family planning could be initiated by both patients and professionals. As previous studies show that professionals are aware of the challenges for patients in discussing their family planning, professionals could initiate a conversation. Possibly, this feeling of reluctance to talk about family planning is less prevalent amongst younger generations, as shown by our respondents <40 years of age, who more often discussed family planning than respondents >40 years of age.

### 4.3. Strengths and Limitations

Our study provides self-reported data from a large sample of (former) patients with mental health problems and a smaller sample of close ones. Although the sample of close ones is limited, the sample is valuable as close ones are often not included in experience surveys in mental health care. We invited *all* members of the panel, irrespective of self-reported mental health disorder and/or problems and status of recovery. This increases the generalizability of findings, as mental health problems are known to be difficult to classify, subject to cultural differences, and subject to the classifier’s interpretation [27]. Our sample was diverse in self-reported mental health problems.

Key limitations are the homogeneity of the sample regarding female gender, educational level, and respondents being middle-aged. There may be a sample bias as panel members might have a more intrinsic motivation to participate in research than the general population of people with mental health problems. Furthermore, non-response bias is a common problem in voluntary recruitment in public health studies and could lead to an underestimation of the severity of the problem, as more healthy persons tend to participate in surveys [28,29]. It is possible that (former) patients and close ones who find it difficult to discuss these topics did not fill out the questionnaires, pointing towards an underestimation of perceived taboo in the general population of people with mental health problems. The response rate of 9% among a panel of persons who regularly respond to questionnaires about mental health was lower compared to another survey amongst the panel (21.5% response rate) [30]. This might indicate that wishes for children and family planning are sensitive topics to address. Additionally, questions regarding motherhood and pregnancy might be less relatable to people of non-female gender, people without a life partner, people with no history of pregnancies, or people who have never experienced challenges with reproductive health. With the broad age range of the respondents, recall bias might influence our results as we inquire about experiences that, in some cases, occurred years ago. Moreover, experiences from decades ago may not represent the current situation in mental health care or reproductive care. As younger respondents had a conversation about family planning more often than older ones, a shift towards discussing family planning in mental health care might already be underway.

### 4.4. Recommendations for Future Research

The results of this exploratory study point to a compelling need to validate the findings in other samples of persons with mental health problems, for example, samples from other geographical regions or inpatients with mental health problems. This would clarify whether our findings are context-specific or universally applicable to persons with mental health problems. Another promising line of research would be to explore the perspective of adolescents and young adults with mental health problems on family planning, as young persons might encounter struggles with family planning in the future. Their experiences might better reflect current practices in mental health care.

Although unintended pregnancies are a global public health problem and evidence of the magnitude of this problem is much needed [31], we also emphasize the need to investigate (involuntary) childlessness and fear of impaired motherhood on an individual level, as we found that individuals with mental health problems struggle with family planning. Future (qualitative) studies should focus on how, when, and by whom family planning and parenting should be addressed in people with mental health problems and their close ones and what information a conversation about family planning should entail addressing the unmet need.

## 5. Conclusions

This study has uncovered the severe and adverse impact of mental health problems on four domains of reproductive health and family planning: reproductive history (including unintended pregnancies), reproductive decision-making, parenting, and sexuality.

Apart from focusing on the prevention of unintended pregnancies, family planning tailored to the needs of those with mental health problems should specifically address (involuntary) childlessness, insecurities about (possible) motherhood, and the influence of mental health on sexuality. It should also be considered that patients experience taboos on discussing sexuality and mental health in general. Ultimately, only one-fifth of the respondents had a conversation about family planning when it was desired, indicating that patients have an unmet need for talking about their challenges (with mental health professionals). Aside from scientific merits, we believe our findings eventually are also important for policymakers. Family planning should be a topic of discussion in psychiatric healthcare.

## Figures and Tables

**Figure 1 ijerph-20-03070-f001:**
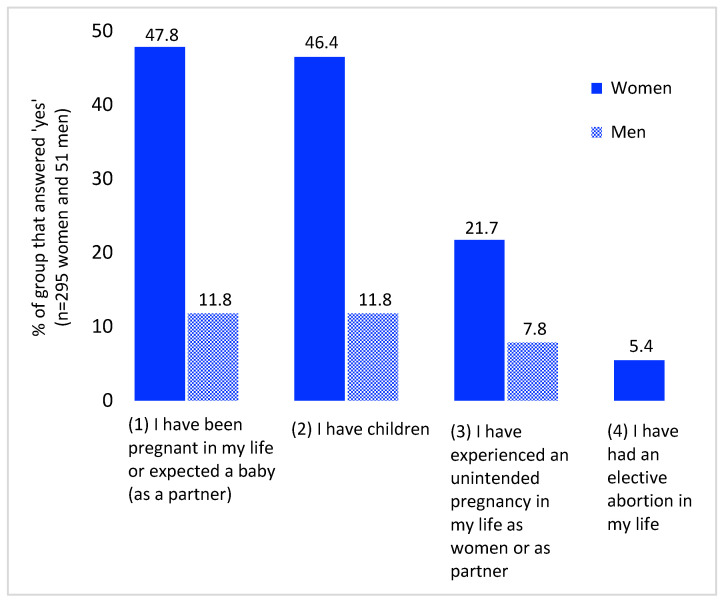
(Former) patients’ responses to questions about reproductive history according to gender (questions 1–4).

**Figure 2 ijerph-20-03070-f002:**
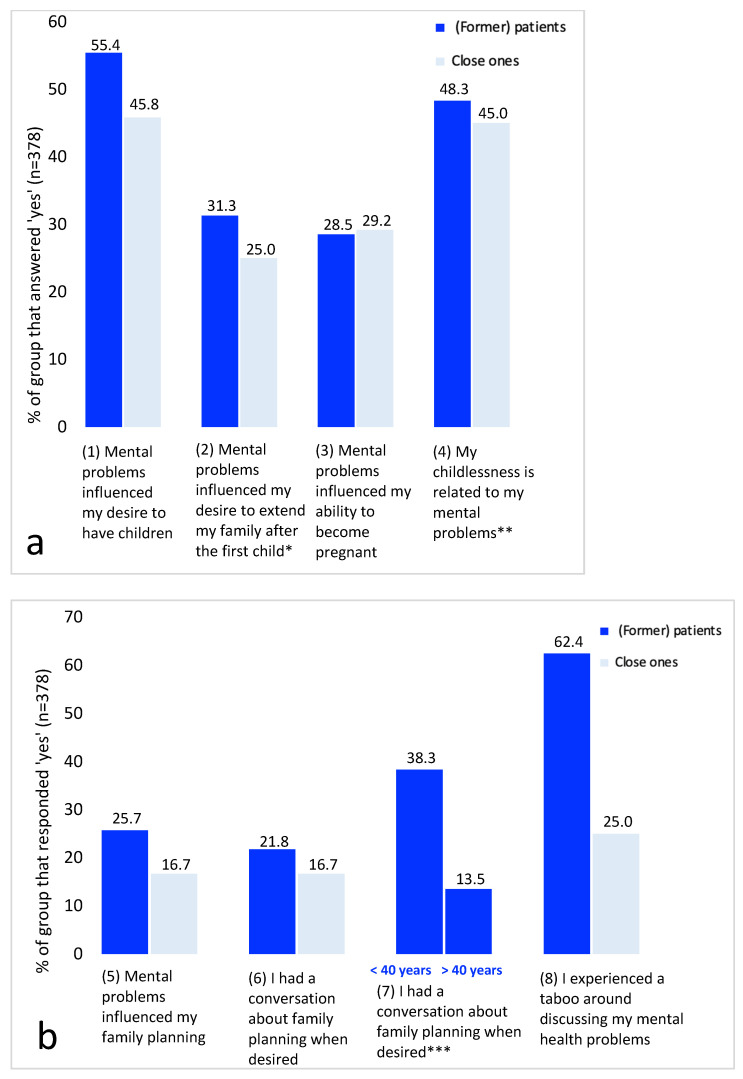
((**a**), questions 1–4) and ((**b**), questions 5–8) Panel’s responses to questions about mental health problems in relation to reproductive decision-making and experiencing taboo. * Proportion of respondents with children (*n* = 151). ** Proportion of respondents without children (*n* = 227). *** Proportions of (former) patients with age <40 years (*n* = 107) and age >40 years (*n* = 235).

**Figure 3 ijerph-20-03070-f003:**
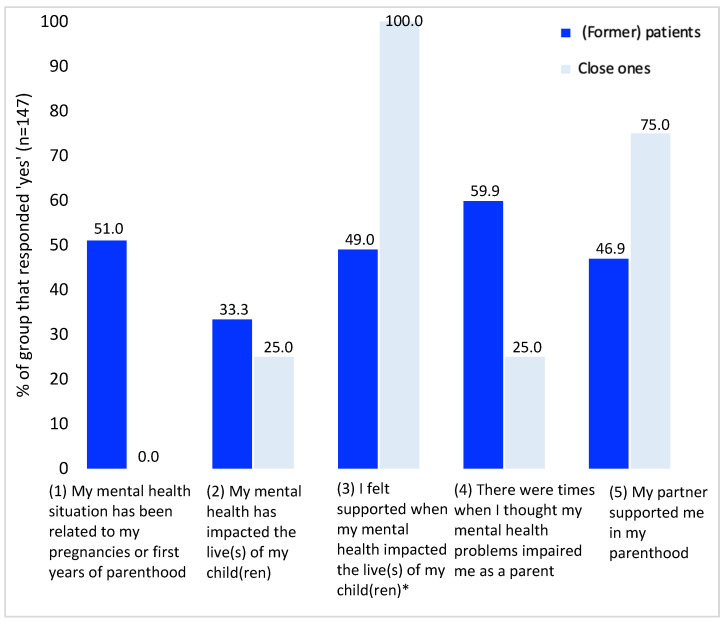
Panel’s responses to questions regarding mental health and parenting (questions 1–5). * Results for respondents who responded ‘yes’ to question (2), *n* = 50.

**Figure 4 ijerph-20-03070-f004:**
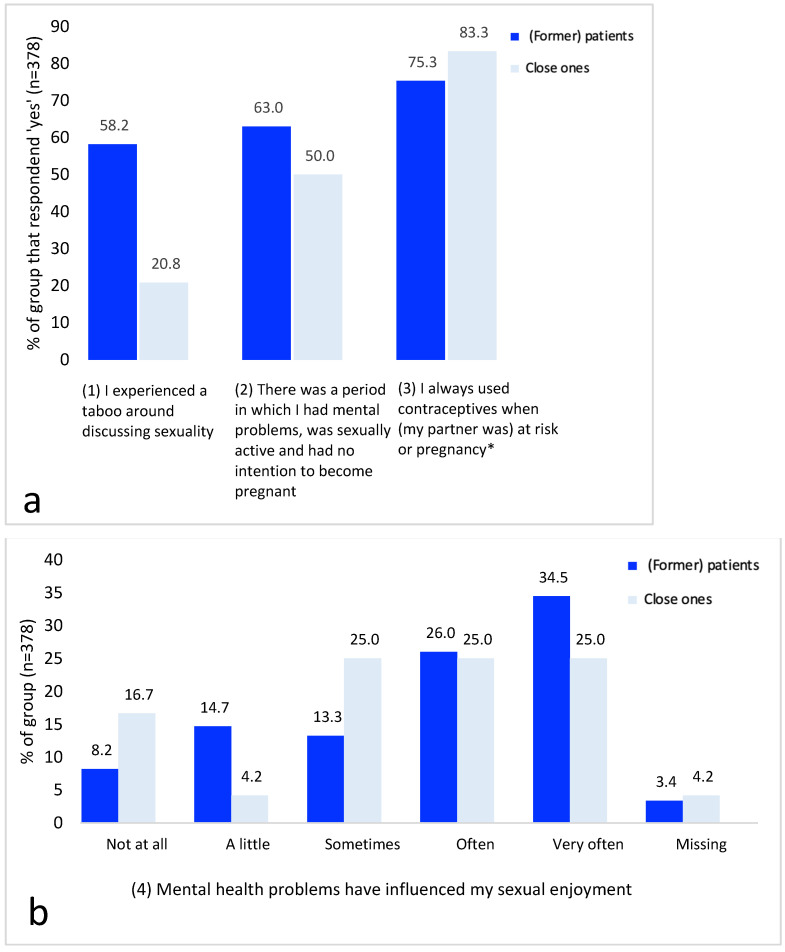
((**a**), questions 1–3) and ((**b**), question 4) Panel**’**s responses to questions regarding mental health and sexuality. * Results for 223 (former) patients and 12 close ones who responded ‘yes’ to question (2).

**Table 1 ijerph-20-03070-t001:** Demographics of survey respondents.

	Total *n* = 378	(Former) Patients *n* = 354	Close Ones *n* = 24
Age (mean; (sd))	47.5 (12.9)	47.7 (12.7)	48.4 (16.2)
Minimum age	20	20	23
Maximum age	84	84	73
Gender (*n* = 374)			
Man	63 (16.7%)	51 (14.4%)	12 (50.0%)
Woman	307 (81.2%)	295 (83.3%)	12 (50.0%)
Other	4 (1.1%)	4 (1.1%)	0
Living area			
Rural	167 (44.2%)	160 (45.2%)	7 (29.2%)
Urban	206 (54.5%)	189 (53.4%)	17 (70.8%)
Missing	5 (1.3%)	5 (1.4%)	0
Education			
ISCED level 6–8 ^1^	190 (50.3%)	178 (50.3%)	12 (50.0%)
ISCED level < 6	184 (48.7%)	172 (48.6%)	12 (50.0%)
Missing	4 (1.0%)	4 (1.1%)	0
Paid job			
Yes	142 (37.6%)	135 (38.1%)	7 (29.1%)
No	227 (60.0%)	211 (59.6%)	16 (66.7%)
Missing	9 (2.4%)	8 (2.3%)	1 (4.2)
Mental health disorder			
Yes	361 (95.5%)	338 (95.5%)	23 (95.8%)^2^
No	16 (4.2%)	15 (4.2%)	1 (4.2%)
Missing	1 (0.3%)	1 (0.3%)	0
Recovered			
Yes	32 (8.5%)	29 (8.2%)	3 (12.5%)
No	158 (41.8%)	147 (41.5%)	11 (45.8%)
I learned to live with it	129 (34.1%)	123 (34.7%)	6 (25.0%)
Other	58 (15.3%)	54 (15.3%)	4 (16.7%)
Missing	1 (0.3%)	1 (0.3%)	0
Medication usage when experienced mental health problems			
Yes	340 (89.9%)	321 (90.7%)	19 (79.2%)
No	38 (10.1%)	33 (9.3%)	5 (20.8%)
Missing	0	0	0
Top 5 self-reported mental health problems in history ^2,3^			
Depression (any type)	222 (58.7%)	216 (61.0%)	6 (25.0%)
Anxiety including OCD	111 (29.4%)	107 (30.2%)	4 (16.7%)
PTSD, Trauma, and stress-disorder	111 (29.4%)	108 (30.5%)	3 (12.5%)
Personality disorder	56 (14.8%)	56 (15.8%)	0
Autism	46 (12.2%)	43 (12.1%)	3 (12.5%)

ISCED, International Standard Classification of Education; PTSD, posttraumatic stress disorder; OCD, obsessive-compulsive disorder. ^1^ Finished college or university. ^2^ More than one possible. Data are presented as numbers (proportions of group ‘Total’, ‘(Former) patients’ or ‘Close ones’). ^3^ Close ones answered this question for mental health disorders of the (former) patients.

## Data Availability

Data are available on request.

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
