# Peer review of "Experiences with Family Planning amongst Persons with Mental Health Problems: A Nationwide Patient Survey"

_ijerph, 2023, doi:10.3390/ijerph20043070_

Round 1

Reviewer 1 Report

I have the following comments.

1. In the demographics of survey respondents the minimum age and maximum age are very high. How much relevant it is?

2. The education lower and higher have proper quantitative demarcation.

3. The other parameters should also be looked into.

4. How much the study is applicable to other regions of the world? A comparative study should be done.

5. The English language should be improved. 

Author Response

Response to Reviewer 1 Comments

I have the following comments.

Response: Thank you for the thorough review of our manuscript and the five questions and/or points for improvement.

Point 1: In the demographics of survey respondents the minimum age and maximum age are very high. How much relevant it is?

Response 1: The broad age range of the respondents has several advantages, but also disadvantages. It is beneficial that older (former) patients and close ones can reflect on their lives and illustrate how mental health problems played a role during the reproductive phase of their life. Moreover, the broad age range enabled us to conduct a subgroup analysis on persons <40 years and ≥40 years (Figure 2b, question 7). Younger persons spoke about family planning more often than older persons. As discussing family planning is a key element of our study, this is a relevant finding. It also illustrates the reviewer’s point: age matters in the scientific discussion about family planning.

We understand the reviewers concern that the age range was wide and overall age high. It is known that young people (especially teenagers) have an increased risk for unintended pregnancies, hence it would have been interesting to include also younger participants. At the same time, new data also shows that older women have an increased risk for unplanned pregnancies (Enthoven, El Marroun et al. 2022). This may be understudied in current literature and our study adds value here.

The survey in the current study was developed for adults. Hence, we included persons ≥18 years only. We addressed the disadvantages of the ‘older’ age of many participants in the limitations section of the discussion paragraph (lines 460-463).

In addition, in the revised manuscript, we now added a recommendation in the discussion section in paragraph 4.3:  

‘Another promising line of research would be to explore the perspective of adolescents and young adults with mental health problems on family planning, as young persons might encounter struggles with family planning in the future. Their experiences might reflect on current practices in mental health care.’ Lines 475-478)

Point 2: The education lower and higher have proper quantitative demarcation.

Response 2: Thank you for this remark. We believe that you mean ‘not proper demarcation’? Assuming this is the case, we have now added the demarcations for education level (according to ISCED) in the methods section and in Table 1 (page 4).

‘Education levels were assessed according to the International Standard Classification of Education (ISCED) levels 0-8.’ (lines 154-155)

Point 3: The other parameters should also be looked into.

Response 3: We apologize, we do not understand exactly to which other parameters you are referring to? Of course, we are very open to give more details on parameters used. Assuming that you would like us to describe the demographics ‘living area’, ‘mental health disorder’, ‘mental health problems’ and ‘recovery status’ in more detail, we have now provided more details on these parameters in the methods section.

‘Mental health disorder was assessed with the question ‘Have you ever received a psychiatric diagnose? Yes/no’. Mental health problems were assessed with the question ‘What kind of mental health problems have you encountered in your life? Open question’. Recovery status was assessed with the question ‘Do you currently consider yourself recovered? Yes/no/I have learned to live with it/other’.’ (lines 156-161)

Point 4: How much the study is applicable to other regions of the world? A comparative study should be done.

Response 4: We thank you for the suggestion and opportunity to improve the recommendations for future research. We acknowledge that our study results are most applicable to the responders of our survey: women of older age and Dutch nationality. Young persons and above all persons from other countries than the Netherlands are currently not represented. In the Netherlands, as in other European countries, unintended pregnancy rates are lower than in other continents, such as Northern America, Africa, and Asia. However, although the magnitude and severity of the problem of unintended pregnancies might differ across geographic regions, we hypothesize that the adverse and severe impact of mental health on family planning, as demonstrated in our study, could be universal. We strongly agree with the reviewer and would stimulate that other studies will replicate our findings. In the revised manuscript, we addressed this by providing recommendation for future research in the discussion section as follows:

‘The results of this exploratory study point to a compelling need to validate the findings in other samples of persons with mental health problems, for example samples from other geographical regions or amongst inpatients with mental health problems. This would clarify whether our findings are context specific or universally applicable to persons with mental health problems.‘ (lines 470-473)

Point 5: The English language should be improved. 

Response 5: To ensure that the revised manuscript has better quality of writing, we included a native speaker in commenting on the final version of the manuscript. This team member has thoroughly read the manuscript and suggested alterations. All changes have been tracked throughout the manuscript to demonstrate these alterations.

Reviewer 2 Report

-        Title: The causal chain is not clear: The reader does not know whether you investigate (unintended) pregnancies among mentally ill people or whether mental health is the outcome of family planning.

-        Title: Mention the study design.

-        Overall, the introduction is very short. More emphasis needs to be spent on describing the rationale for this study.

-        It is a very simple descriptive study. Are there any further stratifications of results possible?

Author Response

Response to Reviewer 2 Comments

Thank you for the review of our manuscript and points to improve it.

Point 1:  Title: The causal chain is not clear: The reader does not know whether you investigate (unintended) pregnancies among mentally ill people or whether mental health is the outcome of family planning.

Point 2: Title: Mention the study design. 

Response 1 &2: Thank you for the suggestion, we agree and have changed the title of the manuscript to reflect the study design and causal chain to: ‘Experiences with family planning amongst persons with mental health problems: a nationwide patient survey’. 

Point 3: Overall, the introduction is very short. More emphasis needs to be spent on describing the rationale for this study. 

Response 3: We now have added several sentences to help clarify our rationale. We hope the introduction has significantly improved. As we have also changed the English writing profoundly in this section, it is challenging to present the improved sentences one by one in this letter. Please find all our alterations in the introduction in the attached revised manuscript.

Below we present several important new or improved sentences that put emphasis on the rationale for the study: lines 46-107

Introduction:

Family planning is an important aspect for most people at some point in their lives [1].

                                               …

Researchers have become increasingly interested in studying the risk factors for impaired family planning, as unwanted pregnancies have adverse impacts on maternal and child health [2]. One of these risk factors is impaired mental health, as, mental health problems can coincide with several aspects of family planning.

                                               …

Although impaired mental health is an established risk factor for unintended pregnancies [4], the perspective and preferences of persons with mental health problems on their own family planning are currently lacking. Moreover, most studies have a profound focus on preventing unwanted pregnancies and only a minor interest in studying how desired pregnancies can be achieved. As the goal of family planning is to have reproductive intentions met, involuntary childlessness amongst persons with mental health problems should be included in the study of family planning. Several factors related to mental health problems could interfere with achieving desired pregnancies, such as problems with sexual functioning [11], concerns of passing on heritable psychiatric conditions [12] and fears of ‘not being a good parent’ [13]. Previous literature has shown that persons with mental illnesses (schizophrenia, autism, eating disorders, substance abuse and/or depression) have lower fecundity when compared to their unaffected siblings [3, 14]. These population-based studies clearly suggest that persons with mental health problems might struggle to have their reproductive intentions met.

Despite these additional challenges facing mental health patients in relation to family planning, their perspectives on childlessness have received little scientific attention.

                                               …

In this study we aimed to elucidate the perspective on family planning from patients with mental health problems and their close ones. We hypothesize that patients and close ones experience various difficulties regarding family planning, and subsequently have a desire to discuss family planning with mental health professionals. Understanding the patients’ perspective will help to tailor family planning counselling to the needs of patients with mental health problems and their close ones.

Point 4: It is a very simple descriptive study. Are there any further stratifications of results possible?

Response 4: We thank you for the suggestion to add further stratifications. Indeed, stratifying the results according to subgroups is insightful. We already stratified for gender, role in the panel ((former) patient or close one) and age <40 or >40 years.

As mentioned in the introduction section, (involuntary) childlessness amongst persons with mental health problems is an understudied topic in science. It has been established that persons with mental health problems have lower fecundity than siblings (Power, Kyaga et al. 2013, Liu, Akimova et al. 2022), but the experiences of patients are lacking. As our sample consists of 207 persons without children, we believe that stratifying the results of childless respondents to various questions yields relevant insights.

In the methods section, we announced these additional analyses:

‘Additionally, we stratified responses for persons with and without children.’ (lines 180-181)

The additional results are integrated in different paragraphs in the results section and the Supplementary Tables 2 and 4. In these two tables, we present all relevant outcomes for (former) patients and close ones who reported to have no children (n=207).

The following sentences have been added to the revised manuscript in the results sections 3.2 and 3.4:

‘Stratifications for (former) patients without children (n=207) showed that mental problems influenced the desire to have children in 75.4%, ability to become pregnant in 36.7%, childlessness in 48.3% and family planning in 24.6% (see Supplementary Table 2). Almost half of all childless (former) patients never had a conversation about family planning (45.9%).’ (lines 265-266)

‘Stratifications for (former) patients without children (n=207) showed that a taboo around discussing sexuality was experienced by 59.9% (see Supplementary Table 4). Sexual enjoyment was influenced by mental health problems ‘often’ in 26.1% and ‘very often’ in 34.8% of the childless (former) patients.’ (lines 338-341)

Reviewer 3 Report

This study aims to explain the perspective on family planning of patients with mental health problems and their closest people. The perspectives on family planning in this study are divided into four domains: reproductive history, reproductive decision-making, parenting, and sexuality. This study has an important contribution to formulating policies to increase understanding of family planning and the prevention of unwanted pregnancies in people with mental health problems.

General concept comments

·         There are too many numbers in the abstract. Authors are advised to avoid showing numbers in the abstract as much as possible.

·         Is the response rate only 9% (line 126)? Please double-check.

·         The figures presented are less attractive. The choice of graphics used makes it difficult for readers to conclude quickly. Authors are advised to use other types of graphics. In addition, including missing and not applicable categories in calculating percentages causes difficulties in making data comparisons between groups of respondents.

·         The conclusion is not in accordance with the data being analyzed. As reviewed in the discussion, the author is suggested to conclude the four perspective domains regarding family planning. The author also has not provided practical and theoretical recommendations for policymakers or proposals for future research.

·         References are still limited, and most of them are old references

Author Response

Response to Reviewer 3 Comments

This study aims to explain the perspective on family planning of patients with mental health problems and their closest people. The perspectives on family planning in this study are divided into four domains: reproductive history, reproductive decision-making, parenting, and sexuality. This study has an important contribution to formulating policies to increase understanding of family planning and the prevention of unwanted pregnancies in people with mental health problems.

Response: Thank you for the evaluation of our manuscript and the proposed improvements.

Point 1:  There are too many numbers in the abstract. Authors are advised to avoid showing numbers in the abstract as much as possible.

Response 1: In line with the suggestion of the reviewer, we removed the numbers from the abstract and have included the key findings as described in the discussion section of the manuscript (lines 489-491).

Point 2:  Is the response rate only 9% (line 126)? Please double-check.

Response 2: The response rate of the panel was 9.1%, as 381 members of 4200 registered panel members responded. However, these panel members can either be active panel members, or be subscribed to the panel for years and never participate in surveys. The MIND organization does not oblige members to respond to surveys. Hence, other MIND surveys have comparable (low) response rates, and this is not specific for this study. The reference to a previously published survey of this panel was included in the manuscript (line 456).

Explanation regarding the response rate is provided in the discussion section (lines 454-456). In addition, based on suggestions by reviewer 4, we paid attention to possible non-response bias in the discussion section (lines 448-451).

Point 3:  The figures presented are less attractive. The choice of graphics used makes it difficult for readers to conclude quickly. Authors are advised to use other types of graphics. In addition, including missing and not applicable categories in calculating percentages causes difficulties in making data comparisons between groups of respondents.

Response 3: In an attempt to make the figures more attractive, we now included the confirmatory category ‘yes’ per question in Figures 1-4. However, we believe that the data from other answer categories (no, not applicable, missing, or different) are informative response categories and are still of utmost importance.

We illustrate this by the following example:

Figure 2, question 5: ‘’Mental problems influenced my family planning’’.

In case we remove the ‘not applicable’ category, and all participants who appointed this option.

  • For (former) patients, the percentage ‘yes’ will change from 25.7% to 36%.
  • For close ones, the percentage ‘yes’ will change from 16.7% to 60% (!) of the participants left in this group.

As demonstrated in the example, the ‘missing’ and ‘not applicable’ answer categories represent large groups of respondents who might 1) feel the question is not applicable to them 2) do not want to answer the question, 3) do not understand the question, etc. We find this information valuable as we aim to understand patients experiences and this information represents (part of) that experience.

To include the information, but also present visually attractive figures at the same time, we have included supplementary tables (Supplementary Tables 1, 2, 3 and 4) with all response categories behind all figures in the revised manuscript. The figures will only present the ‘yes’ category per question.

If desired, readers can read all the detailed information in the Supplementary Tables.

Point 4:  The conclusion is not in accordance with the data being analyzed. As reviewed in the discussion, the author is suggested to conclude the four perspective domains regarding family planning. The author also has not provided practical and theoretical recommendations for policymakers or proposals for future research.

Response 4: Thank you for this suggestion. We have now mentioned the four domains in the conclusions’ section accordingly:

‘This study has uncovered the severe and adverse impact of mental health problems on four domains of reproductive health and family planning: reproductive history (including unintended pregnancies), reproductive decision making, parenting and sexuality.’ (lines 489-491)

In addition, we added a sentence on the theoretical and practical relevance of our findings, in the discussion section:

‘Aside from scientific merits, we believe our findings eventually are also important for policy makers. Family planning should be a topic of discussion in psychiatric healthcare.’ (lines 501-503)

Some proposals for future research were already noted in the original manuscript, we changed the name of paragraph 4.3 into ‘Recommendations for future research’ to clarify the meaning of this paragraph in the revised manuscript (line 468).

We suggested that the perspective of young persons with mental health problems and/or persons with mental health problems in other geographical regions could be studied in the future, as noted by reviewers 1 and 2. These recommendations can be found in the discussion section, not in the conclusions section, as this was suggested by the format of the journal.

Point 5:  References are still limited, and most of them are old references

Response 5: We have added several relevant and/or recent references in the introduction and methods section:

  1. (Hall, Chawla et al. 2023)
  2. (Hall, Benton et al. 2017)
  3. (Power, Kyaga et al. 2013)
  4. (Cheung, Ten Klooster et al. 2017)
  5. (Lindner, Murphy et al. 2001)

Reviewer 4 Report

This paper may be the most complete, thorough, sound research I’ve reviewed in 48 years in higher education! The topic is of much importance. The sample is from a panel who have been involved in research often. While the response rate is unusually low, that fact was recognized by the researchers. I suggest that the authors provide some evidence that the sample is unbiased by non-response. Several methods for examining low response in survey research are available in the literature. See Lindner, J. R., Murphy, T. H., & Briers, G. E. (2001). Handling non-response in social science research. Journal of Agricultural Education, 42(4), 43-53. https://www.jae-online.org/attachments/article/387/42-04-43.pdf for three methods.

I suggest that the authors consider whether using “usage” is better/more appropriate that simply “use.” Generally, I believe that a shorter, more common word is better than a longer, less common word.

I stronglyrecommend publication.

Author Response

Response to Reviewer 4 Comments

This paper may be the most complete, thorough, sound research I’ve reviewed in 48 years in higher education! The topic is of much importance. The sample is from a panel who have been involved in research often. While the response rate is unusually low, that fact was recognized by the researchers.

Response: We are incredibly honored by this compliment. Thank you very much.

Point 1: I suggest that the authors provide some evidence that the sample is unbiased by non-response. Several methods for examining low response in survey research are available in the literature. See Lindner, J. R., Murphy, T. H., & Briers, G. E. (2001). Handling non-response in social science research. Journal of Agricultural Education, 42(4), 43-53. https://www.jae-online.org/attachments/article/387/42-04-43.pdf for three methods.

Response 1: We thank the reviewer for the valuable suggestion to consider the possibility of non-response bias in our sample and for the reference. As the MIND panel is anonymous, we are unable to gain insight in the demographics of the non-responders. What we do know and have mentioned in the discussion section of the revised manuscript: the MIND panel is generally known as a group of persons who are strongly opinionated about mental health. If non-response bias is indeed present (and this is highly likely considering the response rate of 9%), non-responders might have several reasons for not participating. We address these reasons in the limitations section (lines 456-460). 

In addition to the article provided by the reviewer (Lindner, Murphy et al. 2001), a paper specifically from the field of mental health (Cheung, Ten Klooster et al. 2017) on the impact of non-response bias explains that responders usually have better (mental) health.

If this applies to our sample, and we have included persons with better health and more positive experiences, we can conclude that family planning for persons with mental health problems in the general population might be worse. We have added this in the manuscript as follows:

‘Furthermore, non-response bias is a common problem in voluntary recruitment in public health studies and could lead to an underestimation of the severity of the problem, as more healthy persons tend to participate in surveys [28, 29].’ (lines 448-451)

Point 2: I suggest that the authors consider whether using “usage” is better/more appropriate that simply “use.” Generally, I believe that a shorter, more common word is better than a longer, less common word.

Response 2: We have changed the word ‘usage’ to ‘use’ in the manuscript.

I strongly recommend publication.

Round 2

Reviewer 1 Report

Very nicely incorporated all the suggestions.

But some statistical analysis can be included regarding the respondents.

It will also create a lot of interest in the readers. 

Author Response

Response to Reviewer 1 Comments

Very nicely incorporated all the suggestions.

But some statistical analysis can be included regarding the respondents.

It will also create a lot of interest in the readers.

Response:

We thank you for evaluating our improved manuscript.

You address the possibility of including statistical analyses in the manuscript. We have performed a Chi2 as statistical test in the comparisons presented in Figures 1,2 and 4.  In figure 1, we compared confirmative responses between men and women. In sub analyses of Figures 2 and 4, we compared confirmative responses between (former) patients with and without children.

The results of these statistical analyses are presented in Supplementary Tables 1, 2 and 4.

We consider a comparison of (former) patients to close ones not informative, as the findings in close ones are supportive of the findings in (former) patients instead of a contrast.

To incorporate these analyses in the manuscript, we explain the analyses in the methods section: lines 180-183

‘Additionally, we stratified responses for (former) patients and close ones with and without children. Confirmative responses were compared between men versus women or (former) patients with versus without children by Chi2 tests with a p-value considered significant at <0.05.‘

We present the results in the results section and in Supplementary Tables 2 and 4:

Section 3.1: lines 203-205

Amongst the (former) patients, women had significantly more pregnancies (47.8% versus 11.8%, p<0.001) and children (46.4% versus 11.8%, p<0.001) compared to men (see Figure 1).

Section 3.2: lines 267-272

‘Of the childless respondents (n=207), 48.3% attributed childlessness to their mental health state (see Supplementary Table 2). After stratification for childlessness, (former) patients without children confirmed the influence of their mental health problems on desire to have children (75.4% vs. 27.2%, p<0.05) and the ability to become pregnant (36.7% vs. 17.0%, p<0.05) more often than (former) patients with children.

Mental health problems influenced the desire to have children in 75.4%, ability to become pregnant in 36.7%, childlessness in 48.3% and family planning in 24.6%. Almost half of all childless (former) patients never had a conversation about family planning (45.9%).’

Section 3.4: lines 344-346

‘Stratifications for (former) patients with (n=147) and without children (n=207) showed that a taboo around discussing sexuality was experienced by 59.9% (see Supplementary Table 4). Sexual enjoyment was influenced by mental health problems ‘often’ in 26.1% and ‘very often’ in 34.8% of the childless (former) patients. experiences of taboos and sexual enjoyment were comparable (p=0.506 and p=0.971 respectively) (see Supplementary Table 4).’

Moreover, we have made two small alterations in the manuscript:

  • We dissolved one typo ‘questions` (line 31)
  • We added one missing word ‘address’ (line 39)

Reviewer 2 Report

All comments have adequately been addressed and the quality of the manuscript improved a lot.

Author Response

Response to Reviewer 2 Comments

All comments have adequately been addressed and the quality of the manuscript improved a lot.

Response: We thank you for your feedback and are delighted the manuscript has improved a lot.